# Comparison of St. Thomas II Cardioplegia and Modified Del Nido Cardioplegia in Dogs Undergoing Mitral Valve Surgery

**DOI:** 10.3390/vetsci11050201

**Published:** 2024-05-05

**Authors:** Yuki Onuma, Tomohiko Yoshida, Katsuhiro Matsuura, Yuki Aboshi, Mizuki Hasegawa, Shusaku Yamada, Youta Yaginuma

**Affiliations:** 1VCA Japan Shiraishi Animal Hospital, Sayamadai, 4−33−2, Sayama 350-1304, Saitama, Japan; yuk10numa1994@gmail.com (Y.O.); over_the_top_1987@yahoo.co.jp (Y.A.); shusaku_yamada@hotmail.com (S.Y.); yyp.gyym.myy5@gmail.com (Y.Y.); 2VCA Japan Ozenji Pet Clinic, Ozenjinishi 3-27-1, Asaoku, Kawasaki 215-0017, Kanagawa, Japan; 3Department of Clinical Veterinary Medicine, Obihiro University of Agriculture and Veterinary Medicine, Nishi 2-11, Inada, Obihiro 080-8555, Hokkaido, Japan; 4Department of Small Animal Clinical Sciences, College of Veterinary Medicine, University of Florida, 2015 SW 16th Avenue, Gainesville, FL 32608, USA

**Keywords:** mitral valve surgery, myocardial protection, cardioplegic solution

## Abstract

**Simple Summary:**

Mitral valve repair (MVR) has been a common treatment option for myxomatous mitral valve disease (MMVD). MVR is performed under the cardiopulmonary bypass using cardioplegic solution to prevent myocardial injury. St. Thomas II (ST) fluid that is used to perform myocardial protection and cardiac arrest has been used from the earliest procedures of cardiac surgery. However, this fluid must be administered repeatedly at short intervals, which interrupts the surgical process. Del Nido cardioplegia was developed for the immature myocardium that is weak against reperfusion injury. Del Nido cardioplegia can provide longer arresting time upon each administration, which reduces the number of the doses of cardioplegia. Therefore, the surgical procedure becomes simpler and cardiac arrest time or total operation time is shortened in human medicine. These benefits of DN cardioplegia may contribute to an improved outcome of heart surgery in dogs. However, to the author’s knowledge, no studies to date have reported comparisons with traditional cardioplegia fluid (ST fluid). The present study aimed to compare the use of both ST cardioplegia fluid and DN cardioplegia fluid in dogs with MMVD that underwent MVR.

**Abstract:**

Myocardial protection has become an essential adjunctive procedure in veterinary cardiac surgery. Del Nido cardioplegia is a good alternative to the traditional St. Thomas II (ST) cardioplegia in open heart surgery in humans. This study aims to compare intra- and postoperative results between ST cardioplegia and modified del Nido (mDN) cardioplegia in mitral valve surgery in dogs with myxomatous mitral valve disease (MMVD). This retrospective study was conducted using clinical records of 16 MMVD dogs that underwent either ST or mDN cardioplegia. We measured cardiopulmonary bypass (CPB) time, aortic cross-clamp (ACC) time, total operation time, the number of cardioplegia doses, total amount of cardioplegia, required defibrillations, in-hospital mortality and pre- and one-month postoperative echocardiographic variables. CPB (159.4 ± 16.1 vs. 210.1 ± 34.0 min), ACC (101.4 ± 7.0 vs. 136.0 ± 24.8 min) and total operation time (262.3 ± 13.1 vs. 327.0 ± 45.4 min) were significantly shorter in the mDN group (*p* < 0.05). The number of cardioplegia doses (3.25 ± 0.4 vs. 6.25 ± 1.2) and total amount of cardioplegia (161.3 ± 51.5 vs. 405.0 ± 185.9 mL) in the mDN group were also significantly smaller than the ST group (*p* < 0.05). No difference was observed in the requirement of defibrillation, in-hospital mortality and pre- and postoperative echocardiographic variables. The utilization of mDN cardioplegia was associated with shorter operative time in mitral valve surgery in dogs.

## 1. Introduction

Open heart surgery with aortic cross-clamping is a procedure of choice for degenerative mitral valve disease in dogs [1]. Aortic cross-clamping without cardiac arrest by cardioplegia promotes anaerobic metabolism followed by depleting adenosine triphosphate in myocardium, which results in critical cardiac dysfunction even for just a short time [2]. Therefore, establishing an optimal cardioplegia is one of the indispensable factors for achieving better open-heart surgery results. The main goals of hypothermic cardioplegia are as follows: first, to achieve immediate and sustained electromechanical quiescence; second, to obtain rapid and sustained homogeneous myocardial cooling; third, to reduce myocardial oxygen requirement; and finally, to periodically wash out metabolic inhibitors [3,4,5].

St. Thomas II (ST) cardioplegia has been used from the earliest accounts of cardiac surgery [6]. Kanemoto et al. have widely reported the results of using ST cardioplegia in dogs [7]. However, this must be administered repeatedly at short intervals, which interrupts the surgical process. Del Nido (DN) cardioplegia was initially developed for the immature myocardium that is weak against reperfusion injury in the 1990s. DN cardioplegia can provide longer arresting time by each administration, which reduces the number of the doses of cardioplegia [8]. Therefore, the surgical procedure becomes simpler and aortic clamp time or total operation time is shortened in humans [9,10,11]. These benefits of DN cardioplegia may contribute to an improved outcome of heart surgery in dogs. A similar cardioplegic solution was used in a previous report [12]; however, there was no comparison with the traditional cardioplegia.

This retrospective study reviewed clinical records of 16 patients who underwent mitral valve surgery with ST or mDN cardioplegia, and then compared the utility of both cardioplegia in mitral valve repair (MVR) in dogs with myxomatous mitral valve disease. In Japan, DN cardioplegia solution is not commercially available, and therefore, solutions are compounded in order to achieve an equivalence to the del Nido cardioplegia solution (Nephron Pharmaceuticals Corporation, Columbia, SC, USA).

## 2. Materials and Methods

### 2.1. Animals

This retrospective observational study was conducted at Shiraishi Animal Hospital from March 2020 to May 2021. Dogs which underwent MVR with ST cardioplegia formed the ST group and those with modified DN cardioplegia formed the mDN group. All dogs presented with mitral regurgitation due to myxomatous mitral valve disease (MMVD) and were classified as the American College of Veterinary Internal Medicine stage B2–D based on the American College of Veterinary Internal Medicine criteria. Dogs were excluded from the study if they had any of the following: known clinically important systemic or other organ-related disease that was expected to limit the dog’s life expectancy or had undergone emergency surgery in an unusual procedure. Patients’ age, body weight and surgical parameters included the following: aortic cross-clamp (ACC) time, cardiopulmonary bypass (CPB) time, total operation time, the number of cardioplegia doses, total amount of cardioplegia in mL, the number of patients that required defibrillation (4 J/kg), in-hospital mortality and pre- and one-month postoperative echocardiographic variables including transmitral E-wave and A-wave velocity (E/A), left atrium-to-aortic root ratio (LA/Ao), left ventricular end-diastolic internal diameter normalized for body weight (LVIDDN) and fractional shortening (FS).

### 2.2. Surgery and Cardioplegia

We used Miotecter (Fuso Pharmaceutical Industries, Ltd., Osaka, Japan) as ST cardioplegia. In preparation for mDN cardioplegia, we compounded 500 mL of Miotecter, 6.5 mL of MEYLON Injection 8.4% (Otsuka Pharmaceutical Factory, Inc., Tokushima, Japan), 8 mL of MANNITOL Injection 20% (Yoshindo, Inc., Toyama, Japan), 3.25 mL of LIDOCAINE Injection 2% (Sandoz Pharma K.K. Inc., Tokyo, Japan) and 5 mL of ACD-A solution (Terumo Corporation, Tokyo, Japan). All procedures were conducted by the same surgical team using the previously reported left 5th intercostal thoracostal approach [13]. Dogs were induced with a slow infusion of propofol intravenously and anesthesia was maintained on isoflurane. Vasopressors or inotropic supports were used as necessary. Arterial and venous cannula were inserted into the carotid artery and jugular vein, respectively, and connected to CPB. Aortic cannula is inserted into the aortic root and fixed with an anchor suture. Cardioplegia solution is administered antegradely via the cannula. On CPB with moderate systemic hypothermia (30 to 34 °C), diastolic arrest was achieved by ST or mDN cardioplegia. In both groups, the initial dose of cardioplegia was 20 mL/kg to arrest the heart. Temperatures of both cardioplegia were lowered to 4 °C before the coronary perfusion. Repeated dose of ST and mDN cardioplegia at dosage of 10 mL/kg was given at about 20 and 40 min intervals, respectively. Additional doses were administered intermittently when the electrical activity was observed during cardiac arrest. The compositions of both cardioplegia are shown in Table 1. Both cardioplegia and the original DN cardioplegia contain low calcium, and optimal myocardial protection occurs when cardiac arrest is obtained with low calcium cardioplegic solutions [14].

### 2.3. Statistical Analysis

All statistical analyses were conducted using the R software version 4.0.2. Categorical variables were presented in numbers and percentages of total group and compared using the χ2 test or Fisher’s exact test. Continuous variables were presented as mean ± standard deviation and compared using the independent samples *t*-test. *p*-values < 0.05 were considered significant.

## 3. Results

This study includes 16 dogs which underwent MVR to treat mitral regurgitation (ST group: *n* = 8, mDN group: *n* = 8). A dog that underwent pacemaker implantation following mitral valve repair was excluded from the study. There were no statistical differences in demographic variables among both groups (Table 2).

The intraoperative data of the ST and mDN groups are shown in Table 3. All patients were successfully weaned from CPB. In the mDN group, ACC time, CPB time and total operation time were significantly shorter (*p* = 0.003, *p* = 0.0003 and *p* = 0.001, respectively) than in the ST group. The number of cardioplegia doses were significantly lower in the mDN group (*p* = 0.0002). Total amount of cardioplegia (161.3 ± 51.5 vs. 405.0 ± 185.9 mL) was significantly smaller in the mDN group (*p* = 0.015). There was no significant difference between both groups in the requirement of defibrillation, in-hospital mortality and pre- and postoperative echocardiographic variables (Table 4 and Table 5).

## 4. Discussion

We selected MVR as the chosen treatment for MMVD. Medical management always has risks such as pulmonary edema, arrhythmia and sudden death. Surgical treatment can aim for complete cure and minimize those risks. Therefore, this approach was applied to 16 dogs in this study.

Cardioplegia affects the quality of intraoperative procedure and plays a crucial role in preventing myocardial injury during open heart surgery [10]. In order to administer cardioplegia fluid, surgeons have to stop the procedure once. A longer dose interval of cardioplegia during open heart surgery allows for less interruptions when redosing and shorter CPB and ACC duration, which lead to the benefit of the patient and surgeon [15]. In human medicine, DN cardioplegia was initially developed for the pediatric immature myocardium. Nowadays, the safety and the effectiveness of DN cardioplegia in adult open heart surgery have become clear [16]. According to a previous study, CPB and ACC time were significantly shorter in patients who received DN cardioplegia than those who received conventional cardioplegia. Our study demonstrated intra- and early postoperative results in dogs which underwent MVR with ST and mDN cardioplegia. The use of DN leads to shorter ACC and CPB times, reduces cardioplegia dosage and provides potentially better myocardial protection with a safety profile. This result is similar to the previous studies in humans [9,10,11,17]. Repeated doses of ST and mDN cardioplegia were basically given at every 20 min and 40 min, respectively, after the initial dose. The consensus on the redosing interval of cardioplegia is lacking in humans and dogs. A twenty-minute-interval administration of ST cardioplegia is reported in open heart surgery in dogs [7,18]. Moreover, additional administration was performed occasionally when the myocardial action potential was observed during cardiac arrest. The number of doses of mDN cardioplegia was significantly lower than that of ST cardioplegia, which showed that longer cardiac arrest caused by a single dose was achieved by mDN cardioplegia. Both types of cardioplegia are hyperkalemic solutions that induce immediate cardiac arrest. They have one disadvantage: the intracellular Ca^2+^ overload that finally manifests myocardial dysfunction. During arrest, H^+^ accumulates in cardiomyocytes due to ischemia. H^+^ escapes the cell in exchange for Na^+^ through Na^+^/H^+^ exchanger, then Na^+^/Ca^2+^ exchanger is activated and Ca^2+^ accumulates intracellularly, which results to a Ca2+ influx into the cardiomyocytes. In order to mitigate this series of reactions, there are additives in the cardioplegia solution. Mg^2+^ suppresses Ca^2+^ overload independently because it works as a Ca^2+^ channel antagonist that avoids high intracellular Ca^2+^. Lidocaine in mDN cardioplegia is a Na+ channel blocker; therefore, it restrains Ca^2+^ overload through Na^+^/Ca^2+^ exchanger and minimizes the potential for a Na+ window current to occur, which leads to a more certain suppression of the myocardial action potential during cardiac arrest. This membrane stabilization effect of lidocaine is considered to contribute to reinforcing the arrest by mDN cardioplegia.

A lower frequency of injection leads to the following outcomes: 1. a less interrupted intracardiac procedure, 2. shorter CPB and ACC time, and 3. favorable myocardial protection. The cardioprotective effect of each cardioplegia was assessed by postoperative short-term echocardiography. No significant difference in postoperative FS was observed in the mDN group and the ST group. No case in either groups developed clinically relevant systolic dysfunction. Our result indicated that mDN cardioplegia was safe, effective and noninferior to ST cardioplegia in dogs. Considering the potential benefit of mDN for the patient by shortening CPB and ACC time, mDN is a good alternative choice of ST cardioplegia especially for patients with extensive valve lesions, for whom ACC time is estimated to be prolonged. DN cardioplegia solution is compounded and is not readily available in Japan. These compounded solutions for mDN cardioplegia are easily obtained, cost-effective and shorten overall surgical duration. Moreover, there is minimal difference in handling both types of cardioplegia, such as perfusion procedures, because they fit under the same category. Thus, mDN cardioplegia appears to be widely accepted by many cardiac surgeons using mDN for the first time.

The limitations of this study include its retrospective nature and non-randomized design. The sample size is too small to draw valid conclusions that del Nido cardioplegia has better outcomes than St. Thomas II cardioplegia. Most of the patients who underwent surgery were small-sized dogs. Moreover, no long-term follow-up of myocardial protection assessed by echocardiography was performed. There were also no quantitative measurements such as cardiac troponin I to assess the severity of ischemic damage. Further studies with large populations and longer follow-up durations are needed to support these results.

## 5. Conclusions

The present study investigated the utility of modified del Nido cardioplegia fluid. Utilizing mDN leads to shorter ACC and CPB times, reduces cardioplegia dosage and provides a superior myocardial protection effect. However, further studies with larger populations are needed to support these results because the sample size used herein is too small.

## Figures and Tables

**Table 1 vetsci-11-00201-t001:** Composition of each cardioplegia.

Component	ST Cardioplegia	mDN Cardioplegia
Na^+^ (mEq/L)	120	125.82
K^+^ (mEq/L)	16	24.59
Mg^2+^ (mEq/L)	32	30.27
Ca^2+^ (mEq/L)	2.4	2.27
HCO_3_^−^ (mEq/L)	10	21.75
Cl^−^ (mEq/L)	160	161.21
Mannitol(g/L)	0	3.02
Lidocaine(mg/L)	0	122.98

Abbreviations: ST, St. Thomas II; mDN, modified del Nido.

**Table 2 vetsci-11-00201-t002:** Age and body weight of patients of both groups.

Variable	ST Group	mDN Group	*p*-Value
Age (In years)	10.2 ± 1.4	10 ± 2.3	0.857
Body Weight	4.6 ± 1.6	4.12 ± 2.7	0.395

Data are number or mean ± standard deviation. Abbreviations: ST, St. Thomas II cardioplegia; mDN, modified del Nido cardioplegia.

**Table 3 vetsci-11-00201-t003:** Intraoperative data of both groups.

Variable	ST Group	mDN Group	*p*-Value
ACC time (min)	136.0 ± 24.8	101.4 ± 7.0	0.0003
CPB time (min)	210.1 ± 34.0	159.4 ± 16.1	0.0026
Total operation time (min)	327.0 ± 45.4	262.3 ± 13.1	0.0007
Number of cardioplegia doses (times)	6.2 ± 1.2	3.3 ± 0.4	0.0002
Amount of cardioplegia (mL)	405.0 ± 185.9	161.3 ± 51.5	0.015
Number of patients that required defibrillation (head)	1	0	1
In-hospital mortality (%)	0	0	1

Data are number or mean ± standard deviation. Abbreviations: ACC, aortic cross-clamp; CPB, cardiopulmonary bypass; ST, St. Thomas II cardioplegia; mDN, modified del Nido cardioplegia.

**Table 4 vetsci-11-00201-t004:** Preoperative echocardiographic data of both groups.

Variable	ST Group	mDN Group	*p*-Value
HR (bpm)	143 ± 13	149 ± 28	0.627
E (cm/s)	143 ± 25	125 ± 32	0.505
A (cm/s)	89 ± 27	72 ± 26	0.278
E/A	1.7 ± 0.6	2.0 ± 0.9	0.645
LA/Ao	2.3 ± 0.17	2.34 ± 0.63	0.458
LVIDDN	2.19 ± 0.20	1.97 ± 0.33	0.168
FS	49.7 ± 7.2	56.5 ± 11.6	0.160

Data are number or mean ± standard deviation. Abbreviations: HR, heart rate; E, transmitral peak early diastolic velocity; A, transmitral peak late diastolic velocity; E/A, the ratio of transmitral peak early to late diastolic velocity; LA/Ao, left atrial-to-aortic ratio; LVIDDN, left ventricular end-diastolic internal diameter normalized for body weight; FS, fractional shortening; ST, St. Thomas II cardioplegia; mDN, modified del Nido cardioplegia.

**Table 5 vetsci-11-00201-t005:** Postoperative echocardiographic data of both groups.

Variable	ST Group	mDN Group	*p*-Value
HR (bpm)	125 ± 24	130 ± 26	0.520
E (cm/s)	79 ± 19	87 ± 14	0.130
A (cm/s)	105 ± 26	111 ± 13	0.982
E/A	0.78 ± 0.19	0.79 ± 0.18	0.665
LA/Ao	1.51 ± 0.14	1.51 ± 0.34	0.367
LVIDDN	1.54 ± 0.17	1.34 ± 0.30	0.168
FS (%)	38.8 ± 12.6	38.8 ± 7.8	0.900

Data are number or mean ± standard deviation. HR, heart rate; E, transmitral peak early diastolic velocity; A, transmitral peak late diastolic velocity; E/A, the ratio of transmitral peak early to late diastolic velocity; LA/Ao, left atrial-to-aortic ratio; LVIDDN, left ventricular end-diastolic internal diameter normalized for body weight; FS, fractional shortening; ST, St. Thomas II cardioplegia; mDN, modified del Nido cardioplegia.

## Data Availability

The data are available within the text.

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
