# Peer review of "Comparison of St. Thomas II Cardioplegia and Modified Del Nido Cardioplegia in Dogs Undergoing Mitral Valve Surgery"

_vetsci, 2024, doi:10.3390/vetsci11050201_

Round 1
Reviewer 1 Report
Comments and Suggestions for Authors
Reviewer comments for manuscript Id vetsci-2955310 entitled ‘Comparison of St. Thomas II cardioplegia and modified del Nido cardioplegia in mitral valve surgery in dogs’
General comments
Mitral valve disease in dogs is quite common in dogs and primarily managed by medicines that just prolong the life of the patient but complete recovery is rare. Open heart surgery is the preferred choice but is done in limited institutions due to the paucity of expertise, equipment and materials. However, the results are encouraging and total cure cure of the ailment has been frequently reported in literature. The present study is a nice addition to the frequent report of open heart surgery for mitral valve disease in dogs. The manuscript is written flawlessly and I must congratulate the authors that I was not able to point any single mistake. However, I have some concerns with the study. The sample size is too less to draw valid conclusions as the post operative recovery and survival times have not been studied. Moreover, the analysis of biomarkers in the pre and post operative period could have added more validity and strength to the study. It is a common knowledge among veterinary cardiologists that del Nido cardioplegia has better outcomes than St ThomasII cardioplegia. I appreciate the attempt of the authors to investigate an important and critical disease in dogs.
Author Response
Response
We really appreciate the reviewer for understanding the strength of our research and giving us constructive feedback. The reviewer's comments are insightful, which have helped us to improve the manuscript. According to the suggestions from the reviewer, we added the sentence in limitation section (Line 224-225).

Reviewer 2 Report
Comments and Suggestions for Authors
Thank you for performing this work. It is helpful to understand that Del Nido cardioplegia is equivalent to or potentially better than St. Thomas' II. I do agree that a larger prospective study that includes comparison of cardiac biomarkers would be an important follow-up to this work.
Why was a 40 minute dose interval chosen for the del Nido group? It seems fairly well established that dose intervals of up to 90 minutes are safe and effective. Given the ACC time for the del Nido group, it could be possible to use a single dose and even further reduce CPB/ACC times.
For readers that do not regularly perform open heart surgery, it may be helpful to describe what happens during cardioplegia dosing, and how/why frequent dosing prolongs surgery times.
In table 3, defibrillation is reported as a percent, is this percent of patients that required defibrillation? If so, it may be easier and more clear to just report the number of patients since it is a small sample size.
Was any consideration given to evaluation of new post-operative arrhythmia in either group? New arrhythmia (notably atrial fibrillation) is a relatively common finding in people after MVR, it may be interesting to know if the type of cardioplegia used has any impact on development of arrhythmia.
I have also included a highlighted/commented version of the PDF for author review, mostly highlighting grammatical suggestions.

The quality of English is fair and variable throughout the manuscript. In some sections, the English is polished and reads well, and in others, the grammar and word choice make it a little challenging to read and comprehend. It seems that different authors with variable command of the English language may have written different sections of the paper.
Author Response
- Reviewer’s comment
Thank you for performing this work. It is helpful to understand that Del Nido cardioplegia is equivalent to or potentially better than St. Thomas' II. I do agree that a larger prospective study that includes comparison of cardiac biomarkers would be an important follow-up to this work.
â—‹Response
Thank you very much for providing important insights. We are delighted to hear that you think our work will spark debate in our field. In the following, you will find our responses to each of your points and suggestions. We are grateful for the time and energy you expended on our behalf.
- Reviewer’s comment
Why was a 40 minute dose interval chosen for the del Nido group? It seems fairly well established that dose intervals of up to 90 minutes are safe and effective. Given the ACC time for the del Nido group, it could be possible to use a single dose and even further reduce CPB/ACC times.
â—‹Response
In past reports, there were some papers that a perfusionist added cardioplegia fluid every 10 or 20 minutes 1, 2. The appropriate timing to administer cardioplegia fluid is unknown in small animal field. In this study, doses were administered at 40-minute intervals. Based on my experience, I feel that 90 minute intervals are long.
- Mihara K, Kanemoto I, Ando T, et al. Simultaneous surgical repair of a cardiac myxoma causing left ventricular outflow tract obstruction and a ventricular septal defect in a small dog. Open Vet J. 2024;14(2):743-749. doi:10.5455/OVJ.2024.v14.i2.15
- Kurogochi K, Uechi M. Blood cardioplegia reduces intraoperative ventricular fibrillation and transfusion requirements compared to crystalloid cardioplegia in canine mitral valve repair. Am J Vet Res. Published online April 16, 2024. doi:10.2460/ajvr.24.01.0017
- Reviewer’s comment
For readers that do not regularly perform open heart surgery, it may be helpful to describe what happens during cardioplegia dosing, and how/why frequent dosing prolongs surgery times.
â—‹Response
We provided the information about why a longer dose interval results in shorter ACC/CPB times (Line 167-170).
- Reviewer’s comment
In table 3, defibrillation is reported as a percent, is this percent of patients that required defibrillation? If so, it may be easier and more clear to just report the number of patients since it is a small sample size.
â—‹Response
We revised the table 3.
- Reviewer’s comment
Was any consideration given to evaluation of new post-operative arrhythmia in either group? New arrhythmia (notably atrial fibrillation) is a relatively common finding in people after MVR, it may be interesting to know if the type of cardioplegia used has any impact on development of arrhythmia.
â—‹Response
We performed 24-hour ECG monitoring in all cases, and only one case developed atrial fibrillation. Thus, we couldn't conclude if the type of cardioplegia used has any impact on development of arrhythmia in this study.
- Reviewer’s comment
I have also included a highlighted/commented version of the PDF for author review, mostly highlighting grammatical suggestions.
â—‹Response
We would like to express my gratitude to the reviewer for helpful comments. We confirmed the PDF and revised the manuscript.

Reviewer 3 Report
Comments and Suggestions for Authors
The paper submitted by the authors provides pivotal information about mDN cardioplegia compared to St II. Even though It is retrospective study, it shows excellent applications for cardiac intervention. Before acceptance, I have listed some minor comments.
Specific comments:
Line 23, please check "provide".
Line 54, treatment or “procedure”.
Line 79, please delete.
Lines 86-87, it is a little bit redundant.
Line 108, before CPB add the meaning once again.
Lines 123-127, the paragraph is not clear, please redone.
Line 168, please include some explanation of this statement supported by more references.
Lines 176-179, please check.
Line 183, please explain better that about myocardial potential.
In conclusion section, please redo since it doesn’t seem clear. As you explain in the limitations, add some of this here.
Author Response
- Reviewer’s comment
The paper submitted by the authors provides pivotal information about mDN cardioplegia compared to St II. Even though it is retrospective study, it shows excellent applications for cardiac intervention. Before acceptance, I have listed some minor comments.
â—‹Response
We would like to express my gratitude to the reviewer for helpful comments. The reviewer's comments are insightful, which have helped us to improve the manuscript.
<Reviewer’s specific comments>
- Reviewer’s comments
Line 23, please check "provide".
â—‹Response
We revised (Line 23).
- Reviewer’s comments
Line 54, treatment or “procedure”.
â—‹Response
We revised (Line 55).
- Reviewer’s comments
Line 79, please delete.
â—‹Response
We deleted (Line 79).
- Reviewer’s comments
Lines 86-87, it is a little bit redundant.
â—‹Response
We revised (Line 87-90).
- Reviewer’s comments
Line 108, before CPB add the meaning once again.
â—‹Response
We provided the information about cardiopulmonary bypass (CPB) in line 91.
- Reviewer’s comments
Lines 123-127, the paragraph is not clear, please redone.
â—‹Response
Thank you for Reviewer’s comments. We revised the paragraph in line 122- 126.
- Reviewer’s comments
Line 168, please include some explanation of this statement supported by more references.
â—‹Response
We provided the reference in this section [Reference 15].
- Reviewer’s comments
Lines 176-179, please check.
â—‹Response
We revised the sentence (Line 176-179).
- Reviewer’s comments
Line 183, please explain better that about myocardial potential.
â—‹Response
We revised the sentence (Line 183-184).
In conclusion section, please redo since it doesn’t seem clear. As you explain in the limitations, add some of this here.
â—‹Response
We are grateful for reviewer’s kind comment. We revised the conclusion section to take into account limitations (Line 231-234).

Reviewer 4 Report
Comments and Suggestions for Authors
The paper reviews the results of using Saint Thomas vs Del Nido cardioplegia in mitral surgery on dogs. The major finding is that the total procedure times are shorter for Del Nido, which appears to be the result of the increased number of doses of St Thomas that were needed. The other outcomes appear to be unchanged.
1. One small change. The authors should be more specific on their criteria for reapplication of the cardioplegia. Was it completely visual, looking at contractions? Did they measure temperature or electrical activity?
2. The dogs in this study are small (about 10 pounds). Would the authors expect differences in the results if the dogs were larger?
Comments on the Quality of English LanguageGenerally well-written.
Author Response
- Reviewer’s comment
The paper reviews the results of using Saint Thomas vs Del Nido cardioplegia in mitral surgery on dogs. The major finding is that the total procedure times are shorter for Del Nido, which appears to be the result of the increased number of doses of St Thomas that were needed. The other outcomes appear to be unchanged.
â—‹Response
Thank you for giving us the opportunity to strengthen our manuscript with your valuable comments and queries. We have worked hard to incorporate your feedback and hope that these revisions persuade you to accept our submission.
- One small change. The authors should be more specific on their criteria for reapplication of the cardioplegia. Was it completely visual, looking at contractions? Did they measure temperature or electrical activity?
â—‹Response
Cardioplegia were administered intermittently when the electrical activity was observed during cardiac arrest. We provided the additional information about reapplication of the cardioplegia in line 114-115.
- The dogs in this study are small (about 10 pounds). Would the authors expect differences in the results if the dogs were larger?
Thank you for Reviewer’s comments. We think the result doesn’t change. However, it is unclear from this study. Therefore, we added this limitation to manuscript (Line 225).
